# pH-Sensing G Protein-Coupled Receptor OGR1 (GPR68) Expression and Activation Increases in Intestinal Inflammation and Fibrosis

**DOI:** 10.3390/ijms23031419

**Published:** 2022-01-26

**Authors:** Cheryl de Vallière, Jesus Cosin-Roger, Katharina Baebler, Anja Schoepflin, Céline Mamie, Michelle Mollet, Cordelia Schuler, Susan Bengs, Silvia Lang, Michael Scharl, Klaus Seuwen, Pedro A. Ruiz, Martin Hausmann, Gerhard Rogler

**Affiliations:** 1Department of Gastroenterology and Hepatology, University Hospital Zurich, University of Zurich, 8091 Zurich, Switzerland; Cheryl.DeValliere@usz.ch (C.d.V.); jesus.cosin@uv.es (J.C.-R.); katharina.baebler@gmail.com (K.B.); celine.mamie@usz.ch (C.M.); michelle.mollet@gmx.ch (M.M.); cordelia.schuler@gmx.ch (C.S.); susan.bengs@usz.ch (S.B.); silvia.lang@usz.ch (S.L.); michael.scharl@usz.ch (M.S.); pa.ruizcastro@gmail.com (P.A.R.); martin.hausmann@usz.ch (M.H.); 2Sophistolab, 4132 Muttenz, Switzerland; a.schoepflin@sophistolab.ch; 3Zurich Center for Integrative Human Physiology, 8057 Zurich, Switzerland; 4Novartis Institutes for Biomedical Research, 4033 Basel, Switzerland; klaus.seuwen@sfr.fr

**Keywords:** OGR1 (GPR68) expression and function, pH-sensing GPCR, inflammatory bowel disease, fibrosis, fibroblasts

## Abstract

Local extracellular acidification occurs at sites of inflammation. Proton-sensing ovarian cancer G-protein-coupled receptor 1 (OGR1, also known as GPR68) responds to decreases in extracellular pH. Our previous studies show a role for OGR1 in the pathogenesis of mucosal inflammation, suggesting a link between tissue pH and immune responses. Additionally, pH-dependent signalling is associated with the progression of intestinal fibrosis. In this study, we aimed to investigate OGR1 expression and OGR1-mediated signalling in patients with inflammatory bowel disease (IBD). Our results show that OGR1 expression significantly increased in patients with IBD compared to non-IBD patients, as demonstrated by qPCR and immunohistochemistry (IHC). Paired samples from non-inflamed and inflamed intestinal areas of IBD patients showed stronger OGR1 IHC staining in inflamed mucosal segments compared to non-inflamed mucosa. IHC of human surgical samples revealed OGR1 expression in macrophages, granulocytes, endothelial cells, and fibroblasts. OGR1-dependent inositol phosphate (IP) production was significantly increased in CD14+ monocytes from IBD patients compared to healthy subjects. Primary human and murine fibroblasts exhibited OGR1-dependent IP formation, RhoA activation, F-actin, and stress fibre formation upon an acidic pH shift. OGR1 expression and signalling increases with IBD disease activity, suggesting an active role of OGR1 in the pathogenesis of IBD.

## 1. Introduction

The development of an acidic tissue environment is a hallmark feature of a variety of inflammatory processes. Pathologic conditions where localized acidosis may develop include ischemia, metabolic acidosis, cardiac infarction, diabetic ketoacidosis, renal and respiratory failure, malignant tumour growth, and chronic inflammatory diseases, such as rheumatoid arthritis and inflammatory bowel disease (IBD) [1,2]. The association between extracellular tissue acidification and intestinal inflammation is ascribed to increased metabolic demand from infiltrating immune cells, which leads to enhanced glucose consumption and lactic acid formation [3,4,5,6]. Additionally, hypoxia, a common feature in intestinal inflammation, has been shown to decrease the local pH in the mucosal tissue [7]. Consequently, several studies have demonstrated decreased colonic pH in CD and UC patients compared with healthy subjects [8,9,10,11]. During severe inflammation, reduced mucosal bicarbonate secretion, increased bacterial lactate production, and decreased short chain fatty acids might also contribute to the low pH in an IBD patient’s colon [12,13]. To maintain pH homeostasis, it is essential that cells sense changes in extracellular pH and respond accordingly. A family of proton-sensing G-protein-coupled receptors (GPCRs), ovarian cancer G-protein-coupled receptor 1 (OGR1, also known as GPR68), G-protein-coupled receptor 4 (GPR4), and T-cell death-associated gene 8 (TDAG8, also known as GPR65), is activated by protons by differing degrees of sensitivity to pH [14]. Protons bind to histidine residues located at the extracellular region of these transmembrane receptors, activating second messenger signalling pathways, which in turn regulate gene expression [14,15,16,17]. OGR1 is almost silent or inactive at pH 7.8, but fully active at pH ≤ 6.8 [14]. OGR1 couples predominantly through G_q11_ proteins, leading to activation of the phospholipase C (PLC)/inositol phosphate (IP)/Ca^2+^/extracellular signal-regulated kinases (ERK) pathway [14]. However, our previous study also suggests the involvement of the G_α12/13_/Rho signalling pathway [18], which has also been reported by others [19,20,21,22].

Crohn’s disease (CD) and ulcerative colitis (UC), the two main manifestations of IBD, are chronic inflammatory disorders of the gastrointestinal tract. Recently, we and others reported a link between the pH-sensing GPCR family and IBD [18,23,24,25,26,27,28,29,30,31,32]. TDAG8, the anti-inflammatory counter-player to pro-inflammatory OGR1, has been identified as an IBD risk gene by genome-wide association studies [33,34,35,36]. Several studies suggest that TDAG8 negatively regulates inflammation in IBD, supporting the notion of an anti-inflammatory role for TDAG8 [37,38,39]. Our previous studies have shown that IBD patients express higher levels of OGR1 mRNA in the mucosa compared to non-IBD subjects [23,24]. Further, we observed that OGR1 and GPR4 deficiency ameliorates intestinal inflammation in murine experimental colitis [24,32]. We also found that OGR1 is strongly regulated by hypoxia and tumour necrosis factor (TNF) via the nuclear factor (NF)-κB signalling pathway and is essential for intestinal inflammation and fibrosis [23,24,25]. Additionally, we demonstrated that OGR1 blocks autophagy and plays an important role in the regulation of endoplasmic reticulum (ER) stress and the unfolded protein response—an evolutionary mechanism that enables cells to cope with stressful conditions—through the IRE1α-JNK signalling pathway [27]. Finally, in a recent study, we could demonstrate that the pharmacological inhibition of OGR1 ameliorates acute and chronic DSS-induced colitis in mice [40].

Intestinal fibrosis is a common and severe clinical complication of IBD, which frequently leads to surgery in the affected patients [41,42]. Excessive tissue repair following injury promotes fibrosis and is characterized by heightened deposition of extracellular matrix (ECM) proteins. Inflammatory cells, such as macrophages, release factors that stimulate both fibroblast activation and proliferation, resulting in the synthesis and deposition of components of the ECM [43]. In a previous study, we revealed that pH-sensing is implicated in the progression of fibrosis, demonstrating that OGR1 mRNA expression correlates with increased expression levels of pro-fibrotic genes and collagen deposition in paired fibrotic vs. non-fibrotic lesions of terminal ileum obtained from CD patients [25]. Moreover, we observed that OGR1 deficiency in murine models of colitis was associated with a decrease in fibrosis formation [25]. We established that extracellular acidification is linked to intestinal inflammation and leads to fibroblast activation and ECM remodelling [25]. Intestinal fibrosis leads to stricture formation in 50% of CD patients [44,45,46] and 75% of patients with stricturing CD will ultimately require surgery [45,47,48]. In cancer-associated fibroblasts, OGR1 promotes activation of mesenchymal stem cells [49]. Additionally, OGR1 expression increases in pancreatic cancer cells co-cultured with cancer-associated fibroblasts, with subsequent increases in the expression of fibrotic markers in these cells [22].

Our previous studies point to an essential role for OGR1 in the modulation of intestinal inflammation and intestinal fibrosis. However, localization and the cell types responsible for the expression of OGR1 in inflamed and fibrotic tissue of IBD patients remain undescribed due to the lack of suitable antibodies. In the present study, we characterized OGR1 expressing cells in intestinal tissue from IBD patients where OGR1 was detected in monocytes/macrophages, granulocytes, endothelial cells, and fibroblasts. In addition, we evaluated differences in pH-dependent OGR1 activity between monocytes and fibroblasts isolated from peripheral blood samples and the mucosa of IBD patients and healthy controls, respectively. OGR1 signalling significantly increased second messenger IP formation in CD14+ monocytes of IBD patients (active disease) compared to healthy volunteers. We also observed increased OGR1 expression and enhanced signalling in primary human and murine fibroblasts subjected to extracellular acidic pH. Taken together, our data suggest an active role of OGR1 in the pathogenesis of IBD.

## 2. Results

### 2.1. OGR1 mRNA Expression Is Significantly Increased in the Inflamed Mucosa of IBD Patients

Our data show that OGR1 mRNA expression in inflamed colonic resections of CD and UC patients increased five-fold and ten-fold, respectively, compared to mucosal resections from non-IBD control subjects (Figure 1A), thereby confirming our previous results [23,24]. Moreover, OGR1 mRNA expression significantly increased in the inflamed tissue versus non-inflamed tissue from paired samples of IBD patients (Figure 1A).

To further evaluate the relevance of the expression of OGR1 in IBD patients (*n* = 18), we performed a correlation between OGR1 expression and the corresponding clinical score of the patient assigned by the clinician at the time the intestinal resections were taken (Table 1). We observed a significant positive correlation between OGR1 mRNA expression and the clinical score in both the non-inflamed (Figure 1B. r_s_ = 0.5362, *p* = 0.0218 *) and the inflamed mucosa (Figure 1C. r_s_ = 0.6279, *p* = 0.0053 **), supporting the association between OGR1 and intestinal inflammation.

Next, we performed ISH using commercially available probes for human OGR1 on non-inflamed and inflamed tissue from IBD patients and mucosal samples from non-IBD subjects. OGR1 ISH revealed an increased expression in macrophages infiltrating the inflamed tissue compared to non-inflamed tissue (Figure 1D,E). OGR1 ISH also showed OGR1 expression in macrophages (CD68+ cells; Appendix A), endothelial cells (CD31+ cells; Appendix A), and fibroblasts (vimentin+ cells; Appendix A).

### 2.2. OGR1 Immunohistochemistry Shows OGR1 Expression in Healthy Intestinal Mucosa

Recently, Herzig et al. evaluated the OGR1 recombinant rabbit monoclonal antibody (16H23L16) in FFPE normal and neoplastic human tissue samples [50]. We tested this antibody in Caco-2 cells stably overexpressing OGR1. Generation and validation of our stable Caco-2 OGR1 clones U1 and U18 has been previously described [18]. IHC using OGR1 antibody (16H23L16) on OGR1-clone U1 and parental VC cells processed to FFPE cytoblocks revealed strong immunostaining localized to the cytoplasm in OGR1-clone U1 (Figure 2A). VC cells were negative for OGR1 staining (Figure 2B), with only a few cells presenting very slight to almost imperceptible immunostaining. These results are in agreement with qPCR analysis from our previous study, which indicated very low levels of endogenous OGR1 in VC Caco-2 cells [18].

After validating the OGR1 antibody, we next performed IHC on healthy tissue from duodenum and pancreas. In the mucosal duodenum, OGR1-positive cells included macrophages, granulocytes, and endothelial cells (Figure 2C). In the pancreas, OGR1-positive cells (Appendix A) revealed a similar staining pattern to those shown by Herzig [50]. This group elegantly demonstrated that the OGR1-positive cells in the pancreas are glucagon-producing islet cells [50]. In our current study, immunostaining of duodenum tunica muscularis revealed OGR1-positive staining in macrophages, granulocytes, endothelial cells, and fibroblasts (Figure 2D). Additionally, macrophages and endothelial cells were identified in the mucosal duodenum by IHC staining with CD68 (Appendix A) and CD31 (Appendix A), respectively.

### 2.3. OGR1 Protein Expression Is Significantly Increased in the Inflamed Mucosa of IBD Patients

We next performed IHC of colonic resections from non-IBD controls and IBD patients using the anti-OGR1 antibody 16H23L16. Strong immunostaining of OGR1 was observed in all specimens studied (Figure 3A–D and Appendix A), with a stronger OGR1 staining in samples from IBD patients (*n* = 5 CD patients, *n* = 5 UC patients, Figure 3A,B) compared with samples from non-IBD subjects (*n* = 5) (Figure 3C,D). Moreover, the staining was significantly stronger in the inflamed mucosa compared with the paired non-inflamed mucosa of the same patient, confirming that OGR1 expression is higher in active IBD patients (Figure 3A,B). Additionally, macrophages were identified by IHC staining with CD68 (Appendix A and Appendix A).

Further to this, we observed a significant increase in OGR1 protein expression in IBD patients (*n* = 4 CD patients, *n* = 4 UC patients) compared to non-IBD controls (*n* = 5) (Figure 3E,F). Moreover, the inflamed mucosa of both CD and UC patients displayed significantly higher OGR1 protein levels compared with the paired non-inflamed mucosa from the same patient (Figure 3E,F).

### 2.4. Acidic pH Triggers OGR1-Dependent Signalling in Caco-2 Cells Stably Overexpressing OGR1 and Human CD14+ Monocytes

#### 2.4.1. Immunocytochemical Detection of OGR1 in OGR1 Overexpressing Caco-2 Cells

We confirmed the specificity of the OGR1 monoclonal antibody (16H23L16) by immunocytochemical staining in Caco-2 cells stably overexpressing OGR1. The Caco-2 OGR1 clone U1 displayed a strong immunosignal, whereas no visible staining in the parental VC cells was detected (Figure 4A).

#### 2.4.2. OGR1-Dependent Signalling in OGR1 Overexpressing Caco-2 Cells

OGR1 is a G_q_-coupled receptor known to stimulate inositol phosphate (IP) formation upon exposure to slightly acidic pH [14]. We observed significant IP formation in the Caco-2 OGR1 clone U1 compared to Caco-2 VC cells after an acidic pH shift of 30 min (Figure 4B). We previously reported that an acidic environment stimulated reorganization of filamentous actin with marked stress fibre formation in OGR1 overexpressing Caco-2 cells [18]. This suggests that OGR1 couples to G_12/13_, activating the small monomeric GTPase RhoA and other downstream effectors. For this reason, we next investigated RhoA signalling in OGR1 overexpressing Caco-2 cells. Exposure to acidic pH 6.4 for 10 min resulted in activation of RhoA in the OGR1 clone U1, but not in the VC cells (Figure 4C).

#### 2.4.3. Inhibition of OGR1-Dependent Barrier Function with OGR1 Inhibitor (GPR68-I) Using ECIS Technology

Previous studies demonstrate that Rho GTPases are key regulators of F-actin, cytoskeletal dynamics, and barrier function [51,52]. We previously reported that proton activation of OGR1 in Caco-2 cells stably overexpressing OGR1 leads to an acute barrier function enhancement [18]. To confirm the specificity of OGR1 inhibitor (GPR68-I), we subjected Caco-2 clone U1 and VC to acidic pH, with or without the inhibitor (0.5–25 µM), using ECIS technology to monitor barrier function for 24 h. Acidification significantly increased barrier function (resistance) in the OGR1 overexpressing clone U1 (Figure 4D) compared with VC (Appendix A). The inhibitor decreased the resistance in a dose-dependent manner (Figure 4D). The enantiomer of OGR1 inhibitor did not elicit the same effect (Figure 4E).

#### 2.4.4. OGR1-Dependent Signalling in Primary Human Monocytes

Next, we sought to determine whether the increased OGR1 expression observed in active CD and UC patients correlates to OGR1 protein functionality and downstream signalling cascades. IP3, a well-established second messenger, is produced via pH-dependent OGR1 signalling. The formation of IP1, a metabolite of IP3, significantly increased in CD14+ human peripheral blood mononuclear cells (PBMCs) from healthy volunteers and IBD patients upon exposure to extracellular acidic pH for 30 min (Figure 4F,G). Samples in Figure 4F,G were read on the Tecan Infinite or Biotek Synergy instrument, respectively. In the presence of the OGR1 inhibitor (5 to 0.5 µM), IP1 formation decreased in a dose-dependent manner, confirming that pH-stimulated IP1 production is mediated by OGR1 (Figure 4F,G).

### 2.5. Acidic pH Triggers OGR1 Expression and OGR1-Dependent Signalling in Primary Human and Murine Intestinal Fibroblasts

The positive staining for OGR1 observed in fibroblasts in intestinal tissue led us to investigate the pathophysiological relevance of OGR1 in primary human intestinal fibroblasts by IHC using the anti-OGR1 antibody 16H23L16. Moderate immunostaining of OGR1 was observed in fibroblasts treated for 24 h at pH 7.6 and pH 7.4, while cells exposed to acidic pH displayed very strong OGR1 staining (Figure 5A). OGR1 appeared to be mainly cytosolic as no staining was detected in the nucleus. Changes in extracellular pH had no influence on the intracellular localization of OGR1.

#### 2.5.1. OGR1-Dependent Signalling Increases IP Formation in Primary Fibroblasts

Next, we sought to determine whether the increased expression of OGR1 in fibroblasts exposed to acidic pH has an impact on OGR1 protein functionality and downstream signalling cascades by measuring production of IP and RhoA activation, two well-established second messengers downstream of OGR1. We observed significant IP1 formation in primary human fibroblasts (*n* = 6 different human non-IBD fibroblast lines, Figure 5B,C) and primary murine fibroblasts (*n* = 8 different murine fibroblast lines, Figure 5E,F) upon exposure to acidic extracellular pH. Notably, this effect could be reversed in the presence of the specific small molecule OGR1 inhibitor in murine fibroblasts (Figure 5F).

#### 2.5.2. OGR1-Dependent Signalling Leads to Increased RhoA Activation, and F-Actin and Stress Fibre Formation under Acidic Conditions in Primary Fibroblasts

We also observed a significant increase in RhoA activation in primary human and murine fibroblasts upon an acidic pH shift (Figure 5G). RhoA activity was absent in OGR1-deficient murine fibroblasts. In a previous study, we showed that under acidic conditions, OGR1-dependent signalling in Caco-2 cells overexpressing OGR1 stimulated reorganization of cytoskeletal actin with prominent basal stress fibre formation [18]. Therefore, we next performed immunocytochemical staining to examine the status of actin in murine fibroblasts upon exposure to acidic extracellular pH. We observed a striking increase in F-actin stress fibre formation at acidic pH (Figure 5H). Our results indicate an increased activation of RhoA and polymerization of F-actin fibres upon an acidic pH shift for 10 min, which can, in part, be attributed to OGR1.

## 3. Discussion

In this study, we observed OGR1 protein expression in macrophages, granulocytes, endothelial cells, and fibroblasts in human intestinal tissue. Moreover, we found increased expression of OGR1 specifically in inflamed areas of the intestinal tissue from CD and UC patients. Our results show that pH-dependent OGR1/G_q_/PLC/IP activity significantly increased in CD14+ monocytes taken from IBD patients with active disease, or those experiencing disease flares, when compared to CD14+ monocytes from healthy volunteers. In addition, we observed increased OGR1 expression and activity in primary human and murine fibroblasts subjected to extracellular acidic pH. Using genetic and pharmacological approaches, we showed that pH-dependent OGR1 signalling increased IP formation and RhoA activity and induced F-actin stress fibre formation. Earlier work has established that OGR1 can trigger calcium release from intracellular stores, activate protein kinase C, and stimulate serum response factor activity and ER stress [14,18,27,53]. Consequently, we hypothesize that OGR1 activity not only leads to an increased OGR1-mediated pro-inflammatory gene expression but also triggers the activation of signalling cascades involved in further pro-inflammatory responses.

Recently, single-cell RNA sequencing analysis has confirmed the expression of OGR1 in myeloid cells, stromal cells, endothelial cells, and fibroblasts in human intestinal tissue [54,55]. Moreover, it was shown that inflammatory macrophages expressed higher levels of OGR1 mRNA compared to resident macrophages [54]. Likewise, activated fibroblasts expressed higher levels of OGR1 mRNA compared to quiescent fibroblasts [54]. OGR1 mRNA expression has been also described in fibroblasts isolated from other organs, such as human lungs [56] and rat kidneys [57]. Several studies have demonstrated OGR1 expression in cancer-associated fibroblasts [22,58,59]. OGR1 has been shown to promote the activation of mesenchymal stem cells in cancer-associated fibroblasts [49], and OGR1 expression increased in pancreatic cancer cells co-cultured with cancer-associated fibroblasts, enhancing the expression of fibrotic markers in fibroblasts [22]. Increased OGR1 activity in PBMCs suggests a systemic effect rather than one restricted to the intestinal mucosa. We can only speculate that inflammation in the gut might trigger the expression of OGR1 in circulating PBMCs, although further research is needed to confirm this result and to further understand the clinical relevance of this finding.

Links between extracellular acidification, activation of fibroblasts, and ECM remodelling via the activation of pH-sensing GPCR-associated pathways have been described for other organs. Short-term extracellular acidosis leads to pronounced effects on the transcriptional program of rat kidney fibroblasts through the activation of the MAP kinases ERK1/2 and p38, which are involved in the production of the inflammatory mediators TNF, iNOS, and cyclooxygenase 2 [57]. We previously demonstrated that activation of OGR1 by acidic pH increases epithelial barrier function [18]. This effect is linked to a strong induction of F-actin basal stress fibre formation and correlates with the inhibition of epithelial cell migration and proliferation during epithelial wound healing in vitro [18]. OGR1 signalling increased the expression of cytoskeletal, cell adhesion, ECM protein-binding, and inflammatory response genes, as well as several genes linked to ER stress [18]. Using a murine OGR1-deficient model, we demonstrated that acidosis induced the expression of OGR1-dependent genes associated with adhesion, ECM formation, and actin cytoskeletal regulation in WT peritoneal macrophages [24]. Taken together, our previous studies and our new findings strongly suggest that OGR1 is linked to fibrogenesis [18,24,25]. Moreover, OGR1-mediated long-term induction of proteins involved in cell junction assembly and cell/matrix interaction, together with reduced cell motility, highlights an ambivalent role for pro-inflammatory OGR1 in barrier function with overall detrimental effects. The data reported here, together with our previous results [24,40] suggest that the absence of OGR1 has an overall anti-inflammatory effect and a therapeutic use of OGR1 antagonists can be envisaged. However, inhibition of OGR1 may also result in a reduction of wound healing responses and this may preclude a therapeutic use for IBD. Further research is needed to further understand the overall outcome that manipulating OGR1 expression and/or activation might have in IBD patients and under which conditions and in which patient populations OGR1 inhibitors could be used.

Our study has various strengths and limitations. We were able to use a large prospective cohort of well-characterized IBD patients, with access to intestinal tissue samples. Limitations included the small patient sample size for the functional assessment of OGR1, namely OGR1-dependent IP formation. Additionally, the limited number of samples did not allow us to discern differences between the inflammatory and fibrotic patterns present in UC and CD, and further studies with a larger sample size are needed to further investigate these differences. Recruiting patients with active IBD (i.e., experiencing disease flares) willing to give ≥50 mL blood proved problematic. These patients are frequently anaemic and already giving blood for routine tests. Consequently, performing the RhoA activation assay was not possible due to the limited amount of patient material.

In summary, our results indicate a role for the pH-sensing receptor OGR1 in inflammation and fibrogenesis in IBD, thereby providing a potential new target for therapeutic intervention. Further research should be directed towards a better understanding of how acidosis, tissue perfusion, and fibroblast differentiation are influencing inflammation and fibrosis in patients with IBD.

## 4. Materials and Methods

### 4.1. Materials

All chemicals were purchased from Sigma-Aldrich (Buchs, Switzerland), unless otherwise stated. All cell culture reagents were obtained from Gibco (Thermo Fisher Scientific, Reinach, Switzerland), unless otherwise stated. The OGR1 small molecule inhibitor (GPR68-I) and corresponding enantiomer were kindly provided by Takeda Pharmaceuticals, San Diego, CA, USA (Table 2).

Primary antibodies used for immunocytochemistry (ICC): OGR1 (GPR68) recombinant rabbit monoclonal antibody (16H23L16, Invitrogen Cat No. 702595, Thermo Fisher Scientific), anti-actin; and phalloidin for F-actin, conjugated with red-fluorescent BODIPY 558/568 dye (B3475, Invitrogen, Thermo Fisher Scientific, Reinach, Switzerland). Primary antibodies used for immunohistochemistry (IHC): OGR1 (GPR68) recombinant rabbit monoclonal antibody (16H23L16, Invitrogen Cat No. 702595, Thermo Fisher Scientific), cluster of differentiation 68 (CD68) (514H12, Leica Biosystems Cat No. NCL-L-CD68, Biosytems, Muttenz, Switzerland), CD31 (JC70, Cell Marque Cat No. 131M-95, Sigma-Aldrich), vimentin (V9, Diagnostic Biosystems Cat No. Mob090-05, Biosystems), and CD14 (EP128, Epitomics Cat No. AC-0123A, CliniSciences, Nanterre, France). Primary antibodies used for Western blotting: OGR1 custom-made mouse monoclonal developed by AbMart (Shanghai, China).

### 4.2. Human Subjects

Paired surgical resections (non-inflamed and inflamed) were taken from the colon of CD and UC patients within the Swiss IBD cohort study. Non-inflamed colon resections from non-IBD patients undergoing surgery served as our controls. Clinical activity in CD patients was scored using the Harvey–Bradshaw index, which includes parameters such as patient well-being, abdominal pain and resistance, extra-intestinal manifestations, and stool type and frequency. Clinical activity in UC patients was quantified by the MTWSI score, which includes the following parameters: stool frequency/day, diarrhoea, presence of blood in stool, incontinence, abdominal tension and pain, patient status, and medication. The clinical score was blindly assigned by the physician at the time of surgery. The studies were approved by the local ethics committee and all participants signed an informed consent. The participant characteristics are listed in Table 2. Five healthy controls and three IBD patients in flare provided blood samples for IP assays. Demographic and clinical data were obtained at the time of blood collection. Healthy volunteers were recruited as controls.

### 4.3. Animal Models

Animal experiments were performed according to the ARRIVE criteria for in vivo experiments. The generation, breeding, and genotyping of OGR1-deficient (*Ogr1^−/−^*) C57BL/6 mice, initially obtained from Deltagen, Inc., San Mateo, CA, USA, has been previously described [24,53]. Litter mates, C57BL/6 wild type (WT), and *Ogr1^−/−^* mice were used.

### 4.4. Culture of Caco-2 Cells Stably Overexpressing OGR1

Caco-2 cells (LGC Promochem, Molsheim, France) and derived OGR1 overexpressing clones were cultured in 5% CO_2_ at 37 °C in Dulbecco’s modified essential medium (DMEM Glutamax) with geneticin (G418 Sulfate) selective antibiotic (400 µg/mL) and 10% fetal calf serum (FCS, Invitrogen). Construction of the hu-OGR1-pcDNA3.1 + plasmid, OGR1 clone generation, validation, and characterization of the parental vector control (VC) cells, and OGR1 clone U1 and U18 have been previously described [18].

### 4.5. Isolation of CD14+ PBMCs

CD14+ PBMCs were isolated as previously described [31]. In brief, PBMCs were isolated by density gradient centrifugation using Ficoll Histopaque (#10771 Sigma-Aldrich). Cell purification was performed using EasySep Human Monocyte CD14 Enrichment Kit (#17858, Stemcell, Vancouver, BC, Canada) according to the manufacturer’s instructions. The purity of the monocytes was >95% as assessed by fluorescein isothiocyanate-labelled anti-CD14 (557742, BD Biosciences, Allschwil, Switzerland) by flow cytometry.

### 4.6. Isolation and Culture of Human and Murine Intestinal Fibroblasts

Human and murine intestinal fibroblasts were isolated and cultured as previously described [60]. In brief, either endoscopic biopsies or surgical specimens were taken from the designated area of the mucosa. Surgical specimens were cut into 1-mm pieces. Epithelial cells were removed by washing the mucosa in Hank’s Balanced Salt Solution ((HBSS) without Ca^2+^ and Mg^2+^) with 2 mM EDTA, followed by digestion of remaining tissue for 30 min at 37 °C with 1 mg/mL collagenase 1, 0.3 mg/mL DNase I (Boehringer, Mannheim, Germany), and 2 mg/mL hyaluronidase in PBS. Isolated cells were cultured in 25-cm^2^ culture flasks (Costar, Bodenheim, Germany) in DMEM containing 10% FCS, penicillin (100 IE/mL), streptomycin (100 μg/mL), ciprofloxacin (8 μg/mL), gentamycin (50 μg/mL), and amphotericin B (1 μg/mL). Non-adherent cells were removed by subsequent changes of medium. Cells were characterized by immunocytochemistry and used between passages 6 and 12.

### 4.7. pH Modulation

The pH shift experiments were carried out in serum-free RPMI-1640 medium supplemented with 2 mM GlutaMAX and 20 mM HEPES, as previously described [18,24,27,31], unless otherwise stated. For pH adjustment of the medium, the appropriate quantities of NaOH or HCl were added, and the medium was allowed to equilibrate in the 5% CO_2_ incubator at 37 °C for at least 36 h before use. The pH of all solutions was recorded at the beginning of each experiment. All data presented are referenced to pH measured at room temperature. Cells were starved for 4–6 h in serum-free RPMI medium at pH 7.6 to silence pH-sensing receptor OGR1 and then subjected to an acidic pH shift at various pH values.

### 4.8. Immunocytochemistry

Cells were grown in 8-well chamber slides (ibidi, Graefeling, Germany, Cat no. 80826) and subjected to an acidic pH shift, as described above. Cells were fixed with 4% paraformaldehyde at room temperature for 15 min and permeabilized after rinsing with PBS in 100% methanol for 10 min at −20 °C. After blocking in 2% goat serum, 1% bovine serum albumin (BSA), and 0.1% Tween 20, cells were incubated with OGR1 antibody (Cat No. 702595, Invitrogen; dilution 1:1000) overnight at 4 °C, followed by incubation with an Alexa Fluor 488-conjugated anti-rabbit antibody (Cat. No. A11032, Invitrogen; dilution 1:1000) for 1 h and then DAPI for 5 min. Specimens were mounted and images were acquired and processed using the Leica SP5 laser scanning confocal microscope (Leica Microsystems, Wetzlar, Germany) and Leica confocal software (LAS-AF Lite, Leica Microsystems).

### 4.9. Preparation of Cytoblocks

Cells were centrifuged at 400× *g* for 5 min. Four drops of plasma were added to the cell pellet and, after complete mixing, one drop of thrombin (60 IU/mL) was added to the cell suspension. After gentle mixing for 1 to 2 min, the coagulated cell clot was transferred to a CellSafe biopsy capsule and fixed in 4% buffered formalin for at least 10 min. Specimens were processed in the manner of formalin-fixed and paraffin-embedded (FFPE) specimens.

### 4.10. Immunohistochemistry (IHC)

Intestinal specimens were formalin-fixed and paraffin-embedded. Consecutive 3 µm sections were performed using a Leica HistoCore Biocut Rotary Microtome, Biosystems. Immunostaining for OGR1, CD68, CD31, CD14, and vimentin were performed on Leica Bond Max instruments using the BOND Polymer Refine Detection Kit (Leica Biosystems Cat No. DS9800, Biosystems), including all buffer solutions from Leica Biosystems and processed according to the manufacturer’s guidelines. Paraffin slides were dewaxed, pre-treated, and incubated with the designated antibodies: OGR1 (GPR68) antibody (Invitrogen; dilution 1:250, 60 min at RT); CD68 (Leica Biosystems; dilution 1:150, 30 min at RT); CD31 (Cell Marque; dilution 1:200, 30 min at RT); CD14 (Epitomics; 1:200, 60 min at RT), and vimentin (DBS; 1:3000, 30 min at RT).

### 4.11. RNA Isolation and Real-Time Quantitative PCR (qPCR)

Colonic tissue was homogenized using the gentleMACS Dissociator (Miltenyi Biotec, Gladbach, Germany). Total RNA was isolated using the RNeasy Plus Kit (Qiagen, Hombrechtikon, Switzerland) or the Maxwell 16 Total RNA Purification Kit (Promega, Duebendorf, Switzerland) with DNase treatment, according to the manufacturer’s instructions. The High-Capacity cDNA Reverse Transcription Kit (Applied Biosystems, Thermo Fisher Scientific) for reverse transcription was used following the manufacturer’s instructions. qPCR was performed on a Fast HT7900 Real-Time PCR system (Applied Biosystems) under the following PCR program: 20 s at 95 °C, followed by 40 cycles of 95 °C for 1 s and 60 °C for 20 s with the TaqMan FAST Universal Mastermix. Relative mRNA expression was determined by the comparative ΔΔCt method using the reference gene β-actin. All gene expression assays were obtained from Applied Biosystems (human OGR1 (Hs00268858_s1) and human β-actin (4310881E)).

### 4.12. Immunoblotting

Colonic tissue was homogenized in the gentleMACS Octo Dissociator (Milteny Biotec), then lysed in M-PER buffer containing protease inhibitor cocktail. The following antibodies were used: monoclonal OGR1 antibody (Abmart) or β-actin (Merck Millipore, Sigma-Aldrich) at 1:1000 and 1:2500 dilution, respectively. Protein expression was quantified by densitometry using ImageJ 1.44 software. Data were normalized to β-actin.

### 4.13. In Situ Hybridization (ISH)

ISH for OGR1 RNA was performed using RNAscope 2.5 HD assay-Brown (Advanced Cell Diagnostics, Hayward, CA, USA) by Advanced Cell Diagnostics (ACD), according to the manufacturer’s protocol, on FFPE colon tissue using a commercially available OGR1-specific probe (Hs-GPR68, ACD 411211) designed to detect +strand RNA.

### 4.14. IP Formation Assay

Formation of myo-inositol 1 phosphate (IP1), a metabolite of IP3, following H^+^ activation of ORG1 via the G_αq11/_PLC/IP3/Ca^2+^ pathway was measured by a competitive cell-based sandwich immunoassay and quantified by homogeneous time-resolved fluorescence (HTRF) technology (CisBio IP-One, 62IPAPEC, CisBio, Saclay, France) according to the manufacturer’s instructions. Cells were seeded in serum-free HBSS buffer supplemented with 25 mM HEPES in 384-well plates (Cat. No. 781080, Greiner Bio-one, Kremsmuenster, Austria) at non-activating pH 7.6, with or without the OGR1 antagonist (10 µM), and incubated for 15 min, followed by a 30-min pH shift, which was achieved by addition of the appropriate amount of HBSS buffer to obtain the desired final pH (pH 7.6 or pH 6.5). Lithium chloride (50 nM), which prevents the degradation of IP1 to D-myo-inositol, was used in all conditions. All incubations were carried out in a non-CO_2_ incubator at 37 °C. Samples were read on the Tecan Infinite M1000 (Thermo Fischer Scientific) or Biotek Synergy H1 (171124A, BioTek, Winooski, VT, USA) instrument. As the reading of the samples from these two machines is based on differing technologies, the data cannot be combined.

### 4.15. RhoA GTPase Activation Assay

pH-mediated RhoA activity was measured as previously described [31]. Briefly, after 2 h of starvation at pH 7.6 in serum-free RPMI to silence OGR1 receptor, cells were subjected to acidic pH or the non-activating pH (pH 7.6–7.8) for 10 min. All incubations were carried out in a 5% CO_2_ humidified 37 °C incubator. Following treatment, cells were lysed and activated RhoA levels were determined using the RhoA G-LISA Activation Assay kit (#BK124, Cytoskeleton, Denver, CO, USA) according to the manufacturer’s instructions.

### 4.16. Cell Resistance Measurements by Electric Cell-Substrate Impedance Sensing

Resistance and impedance of Caco-2 monolayers were monitored using electric cell-substrate impedance-sensing (ECIS) instruments (Applied BioPhysics, Troy, NY, USA) as previously described [18].

### 4.17. Statistical Analysis

Statistical analyses were performed using GraphPad Prism (v9.0, La Jolla, CA, USA). Data are presented as mean ± SD or mean ± SEM. The unpaired t-test was used to compare differences between two groups. When comparing three or more groups, one-way ANOVA was performed and followed by the post hoc Turkey test. The clinical correlations were analysed using Spearman’s rank coefficient (r_s_). A *p* value of <0.05 was considered to be statistically significant. Throughout this article, asterisks denote significant differences at * *p* < 0.05, ** *p* < 0.01, and *** *p* < 0.001.

## 5. Conclusions

OGR1 expression and activity significantly increase with IBD disease activity in IBD-relevant cell types, suggesting an active role of OGR1 in the pathogenesis of IBD. Consequently, OGR1 expression and/or activity could constitute a valuable biomarker for IBD, as it correlates with disease activity. Moreover, manipulating OGR1 activation could constitute a novel therapeutic approach for IBD.

## Figures and Tables

**Figure 1 ijms-23-01419-f001:**
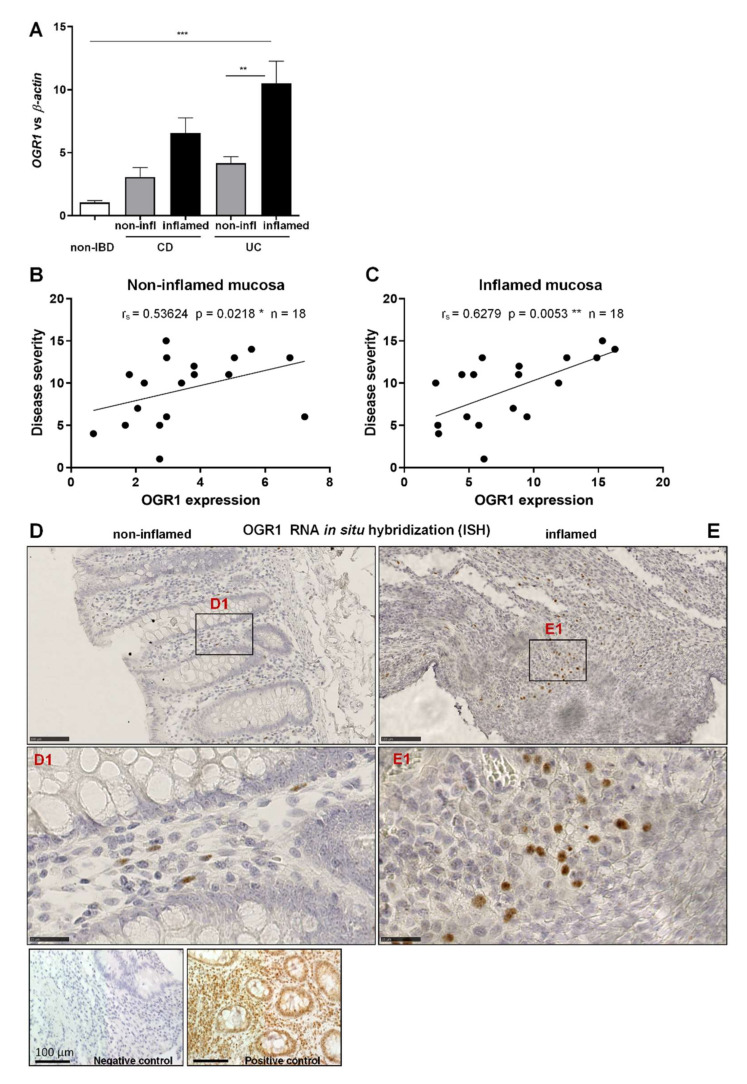
OGR1 mRNA expression is significantly increased in inflamed mucosa of IBD patients. (**A**) OGR1 mRNA expression levels in colonic resections from five non-IBD subjects, and paired samples (non-inflamed and inflamed) from eight CD patient and 11 UC patients. Data are presented as mean ± S.E.M. and statistical analysis was performed using one-way ANOVA. ** *p* < 0.01; *** *p* < 0.001. (**B**,**C**) Correlation of disease severity (Harvey–Bradshaw index for CD patients, Truelove and Witts severity index for UC patients) to OGR1 mRNA expression. Statistical analysis: Spearman’s rank correlation coefficient. (**B**) Non-inflamed mucosa: r_s_ = 0.5364, *p* = 0.0218, *n* = 18. * *p* < 0.05. (**C**) Inflamed mucosa: r_s_ = 0.6279, *p* = 0.0053, *n* = 18. ** *p* < 0.01. (**D**,**E**) In situ hybridization (ISH) identifying OGR1 mRNA (brown dots) in colonic tissue. Representative images are shown from a CD patient. *N* = 2 non-IBD subjects, paired (non-inflamed and inflamed) samples *n* = 2 CD patients, *n* = 2 UC patients. Scale bars: (**D**,**E**) overview images 100 µm; (**D1**,**E1**). Detail images, 25 µm. Controls: negative control, positive RNAscope control (peptidyl-prolyl cis-trans isomerase B). CD: Crohn’s disease; IBD: inflammatory bowel disease; OGR1: ovarian cancer G-protein-coupled receptor 1; UC: ulcerative colitis.

**Figure 2 ijms-23-01419-f002:**
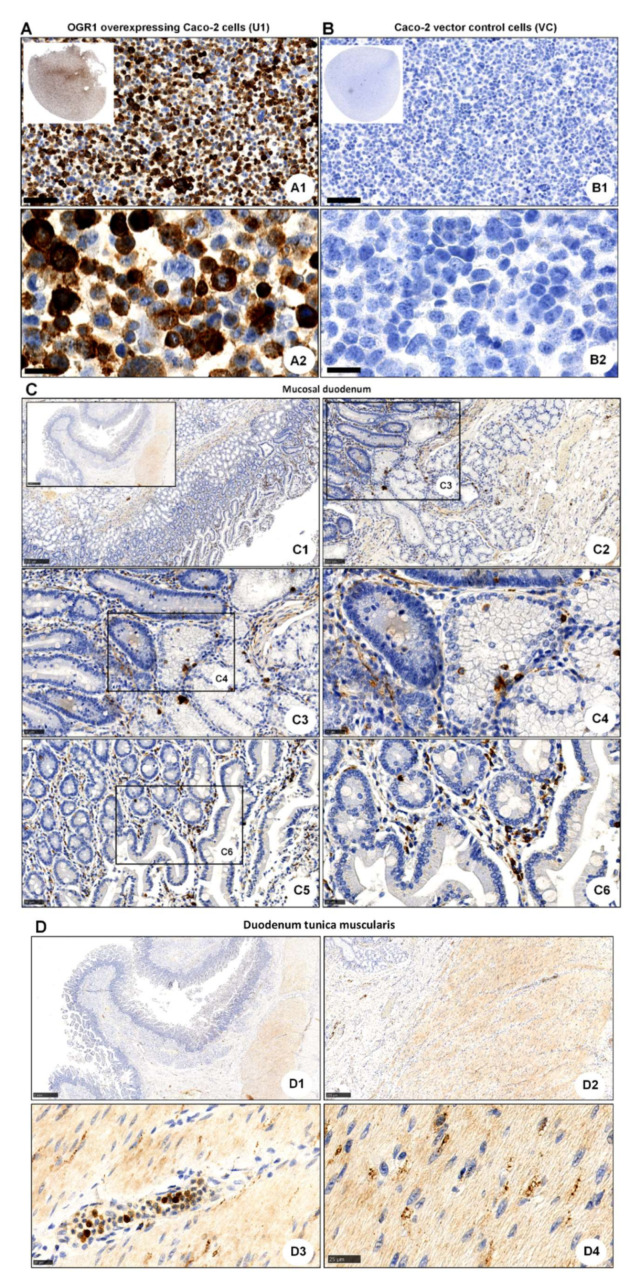
Immunohistochemical detection of OGR1. (**A**,**B**) OGR1 (GPR68) recombinant rabbit monoclonal antibody (16H23L16) for immunohistochemical (IHC) detection of OGR1 was verified in previously validated tools. Caco-2 clone stably overexpressing OGR1 (clone U1); negative control, Caco-2 vector control (VC) cells. (**A1**,**A2**) Strong positive OGR1 staining (brown colour), predominantly in the cytoplasm, is present in OGR1-clone U1. (**B1**,**B2**) No specific staining for OGR1 in VC cells. Counterstaining with haematoxylin. Scale bar: (**A1**,**B1**) 100 µm and (**A2**,**B2**) 25 µm. (**C**) Localisation of OGR1 in healthy human duodenal mucosa. Scale bars: C. Insert overview image 1 mm; detail images, (**C1**) 250 µm; (**C2**) 100 µm; (**C3**,**C5**) 50 µm; (**C4**,**C6**) 25 µm. (**D**) Localisation of OGR1 in duodenal tunica muscularis. Scale bars: (**D1**,**D2**) 1 mm; detail images, (**D3**,**D4**) 25 µm. OGR1: ovarian cancer G-protein-coupled receptor 1.

**Figure 3 ijms-23-01419-f003:**
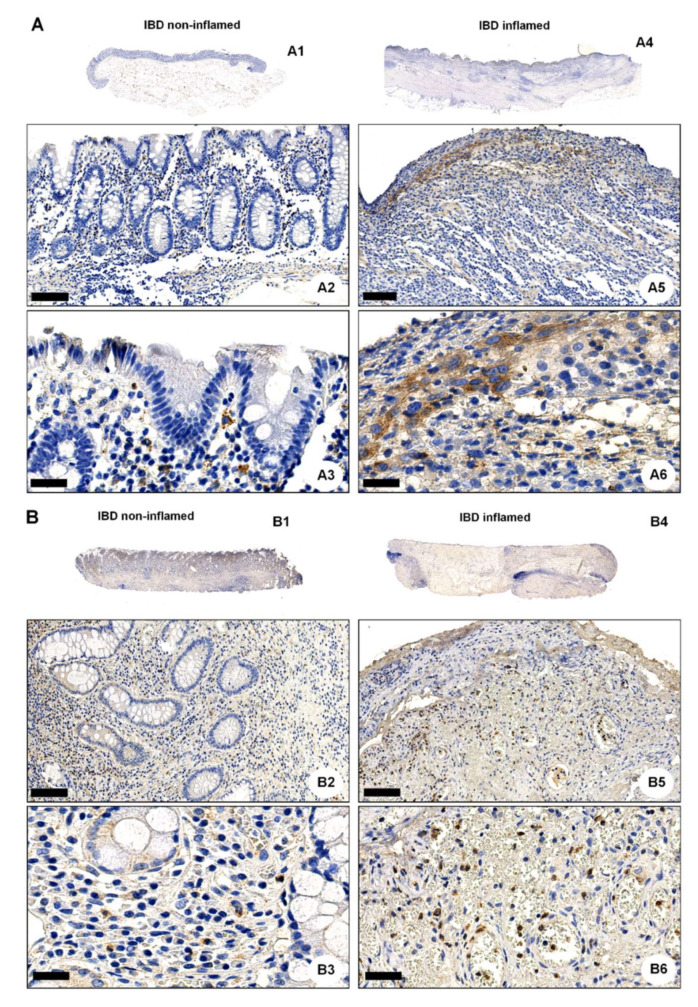
OGR1 protein expression is significantly increased in the inflamed mucosa of IBD patients. Immunohistochemical detection of OGR1 using the anti-OGR1 antibody 16H23L16. Paired samples: (**A1**–**A3**,**B1**–**B3**) non-inflamed; (**A4**–**A6**,**B4**–**B6**) inflamed. Representative images shown are from two CD patients. Scale bars: (**A2**,**B2**) and (**A5**,**B5**) detail images, 100 µm; (**A3**,**B3**) and (**A6**,**B6**) 25 µm. (**C**,**D**) Representative images shown are from two non-IBD subjects. Scale bars: (**C1**,**D1**) 100 µm; (**C2**,**D2**) detail images, 25 µm. Intestinal resections were taken from five non-IBD subjects and paired (non-inflamed and inflamed) samples from five CD patients and five UC patients. (**E**) Total protein was isolated from colonic tissue samples and Western blotting was performed using an OGR1 (GPR68) custom mouse monoclonal antibody (AbMart). (**F**) Densitometry after normalization of OGR1 to β-actin: five non-IBD subjects, four CD patients and four UC patients. Statistical analysis was performed using one-way ANOVA followed by Tukey’s post-test (** *p* < 0.01; *** *p* < 0.001). CD: Crohn’s disease; IBD: inflammatory bowel disease; OGR1: ovarian cancer G-protein-coupled receptor 1; UC: ulcerative colitis.

**Figure 4 ijms-23-01419-f004:**
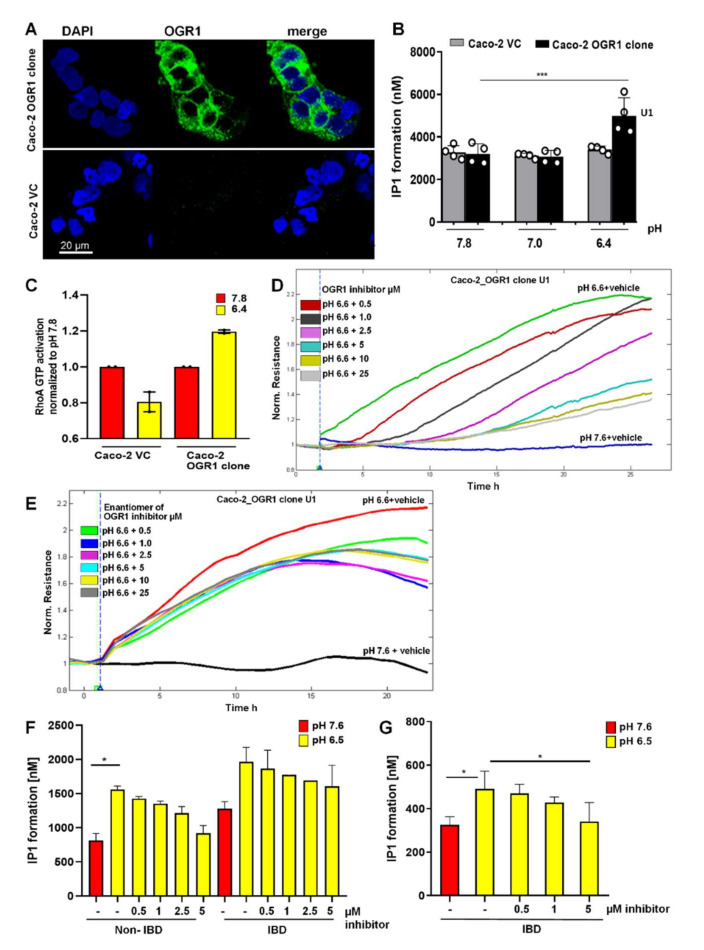
The OGR1 inhibitor blocked OGR1-associated downstream events in Caco-2 cells and human CD14+ monocytes. (**A**) OGR1 (GPR68) antibody 16H23L16 for immunocytochemical detection of OGR1 was verified in previously validated tools. Caco-2 clone stably overexpressing OGR1 (clone U1); negative control, Caco-2 vector control (VC) cells. Cells were examined by confocal microscopy (Leica, Germany). (**B**–**F**) Functional assays confirmed pH-dependent OGR1-mediated signalling. (**B**) U1 and VC cells were subjected to an acidic pH shift and intracellular inositol phosphate (IP) formation was measured. Data are presented as mean ± SD and statistical analysis was performed using one-way ANOVA. *** *p* < 0.001. (**C**) Activated GTPase RhoA was measured in U1 and VC cells upon an acidic pH shift. (**D**,**E**) Efficacy of the OGR1 inhibitor (GPR68-I) and the enantiomer of the inhibitor were measured using electric cell-substrate impedance-sensing (ECIS) technology. OGR1 clone U1 and VC grown to confluent monolayers were subjected to an acidic pH shift with or without the OGR1 inhibitor (0.5, 1.0, 2.5, 5, 10, 25 µM). Changes in resistance of cell monolayers were monitored in real time. Representative graph of eight independent experiments shown for the inhibitor or the enantiomer. (**F**,**G**) Human CD14+ monocytes from healthy volunteers and active IBD patients were subjected to an acidic pH shift, with or without the OGR1 inhibitor (0.5, 1.0, 2.5, 5 µM). IP1 production was measured from healthy volunteers (*n* = 5) and active IBD patients (*n* = 2); active IBD patient (*n* = 1). Samples in (**F**,**G**) were read on the Tecan Infinite or Biotek Synergy instrument, respectively. Data are presented as mean ± SD and statistical analysis was performed using one-way ANOVA. * *p* < 0.05. IBD: inflammatory bowel disease; OGR1: ovarian cancer G-protein-coupled receptor 1; vector control: VC.

**Figure 5 ijms-23-01419-f005:**
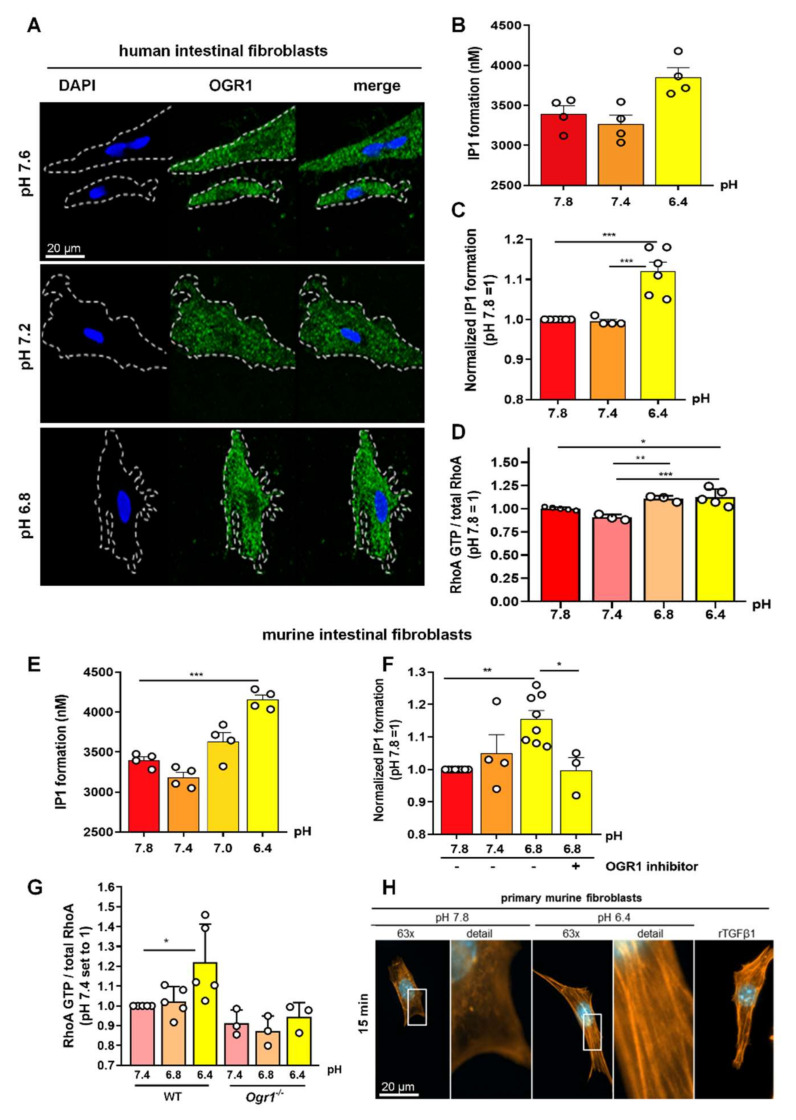
OGR1 expression and OGR1-dependent signalling in primary human and murine intestinal fibroblasts. (**A**–**D**) Primary human fibroblasts. (**A**) Immunocytochemical (ICC) detection of OGR1 was confirmed in human fibroblasts using OGR1 (GPR68) antibody 16H23L16. Functional assays confirmed pH-dependent OGR1-mediated signalling. (**B**,**C**) Fibroblasts were subjected to an acidic pH shift and intracellular inositol phosphate (IP) formation was measured (*n* = 4–6 different fibroblast lines isolated from non-IBD patients). (**D**) Activated GTPase RhoA was measured in fibroblasts after an acidic pH shift (*n* = 3–5 different fibroblast lines isolated from non-IBD patients). (**E**–**H**) Primary murine fibroblasts. (**E**,**F**) Fibroblasts were subjected to an acidic pH shift and IP formation was measured (*n* = 8, different murine fibroblast lines). pH-dependent OGR1-mediated signalling in fibroblasts leads to a significant increase in IP formation and is reversed by an OGR1 inhibitor. (**G**) RhoA activity significantly increases in WT fibroblasts but is absent in OGR1-deficient fibroblasts (*n* = 6 different WT fibroblast lines, *n* = 3 different *Ogr1**^−/−^* fibroblast lines). (**H**) Proton-activated OGR1-mediated signalling in fibroblasts increases filamentous actin (F-actin) stress fibres at acidic pH. Positive control: rTGFβ1. Data are presented as mean ± SEM and statistical analysis was performed using one-way ANOVA. * *p* < 0.05; ** *p* < 0.01; *** *p* < 0.001. IBD: inflammatory bowel disease; OGR1: ovarian cancer G-protein-coupled receptor 1; rTGFβ1: recombinant transforming growth factor beta 1; WT: wild type.

**Table 1 ijms-23-01419-t001:** Participant characteristics.

	Non-IBD	CD	UC
Number of patients	5	8	10
Gender, females	3 (60%)	4 (50%)	4 (40%)
Age (mean ± SD)	55.2 ± 25.8	40.2 ± 10.7	31.8 ± 15.0
Disease severity			
Harvey–Bradshaw index (median, IQR)	NA	6.0	NA
UC severity index (median, IQR)	NA	NA	10.5
Medical history			
Azathioprine/6-MP	NA	5/11 (45.5%)	7/9 (58.3%)
Methotrexate	NA	1/11 (9.1%)	0/9 (0.0%)
Anti-TNF	NA	6/11 (54.5%)	2/9 (16.7%)
Systemic steroids	NA	1/11 (9.1%)	3/9 (25.0%)
NSAID intake	NA	2/11 (18.2%)	6/9 (75.0%)

Abbreviations: 6-MP, mercaptopurine; CD, Crohn’s disease; IBD, inflammatory bowel disease; IQR, interquartile range; NA, not applicable; NSAID, nonsteroidal anti-inflammatory drug; SD, standard deviation; TNF, tumour necrosis factor; UC, ulcerative colitis.

**Table 2 ijms-23-01419-t002:** OGR1 inhibitor (GPR68-I).

Characteristics of Compound GPR68-I	
Mol weight (g/mol)	375.4
Solubility in water (µM)	120
Cytotoxicity (µM)	100
IC50 * Human (nM)	12

* Value was measured in calcium flux assays in recombinant Chinese hamster ovary (CHO)-K1 cells overexpressing human OGR1 upon exposure to acidic pH.

## Data Availability

The data underlying this article will be shared on reasonable request to the corresponding author.

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
