# Peer review of "pH-Sensing G Protein-Coupled Receptor OGR1 (GPR68) Expression and Activation Increases in Intestinal Inflammation and Fibrosis"

_ijms, 2022, doi:10.3390/ijms23031419_

Round 1
Reviewer 1 Report
The present manuscript by de Vallieère and coworkers characterizes the expression of ovarian cancer G-protein coupled receptor (OGR1), a member of the proton sensing G-protein coupled receptor family in intestinal tissue from patients with inflammatory bowel disease (IBD). Main conclusions of the manuscript are that OGR1 expression is higher in IBD patients as compared to controls and higher in inflamed tissue than in non-inflamed tissue specimens and that protein expression is restricted to macrophages/monocytes, granulocytes, endothelial cells and fibroblasts in human intestine. The authors extend their findings and assess OGR1 activity and signaling in cell culture experiments using primary murine and human fibroblasts and PBMCs exposed to an acidic environment. The methods used are adequate, the paper is well written and the work is of clinical importance in the field of IBD. The major strength of the work is that OGR1 expression as well as function is assessed in humans and primary human cell culture assays respectively. Some minor comments arise which may help the readers to better understand the manuscript.
Comments:
The part of intestine, where specimens from IBD and control patients were taken from, needs to be clearly stated in the M&M section (ileum, right/left sided part of the colon). Does gene expression differ in the various part of the intestine? If yes, this must be considered in the results section (Fig 1&3). Where all specimens endoscopic biopsies ?
Is there any evidence that the pH is acidic in intestinal inflammation in IBD?
OGR1 activity may increase “wound healing” and barrier function but may also be detrimental in IBD patients in regard to fibrosis. The reader is somehow left alone with the statement that OGR1 could be a therapeutical target. Would it be wise to inhibit or even stimulate OGR1?
Increased OGR1 activity was detected in PMBCs from IBD patients. This might be somehow unexpected since this would indicate a systemic effect and not a local one restricted to the intestine. The authors might want to add a comment in that regard.
What could be the mechanisms of increased OGR1 activity in response to acidosis in IBD patients? Is this solely increased gene expression? The authors might want to speculate on that.
Author Response
Dear Reviewer,
We thank you for the comments you provided which have highlighted some important areas for improvement of the manuscript. We have considered and addressed all your comments and suggestions, as detailed below. We trust that our responses and the revised version of the manuscript fully satisfy your requests and that our manuscript is found suitable for publication in International Journal of Molecular Sciences.
- The part of intestine, where specimens from IBD and control patients were taken from, needs to be clearly stated in the M&M section (ileum, right/left sided part of the colon). Does gene expression differ in the various part of the intestine? If yes, this must be considered in the results section (Fig 1&3). Where all specimens endoscopic biopsies?
Response: This is indeed an important point that needed to be addressed. We used colonic resections from IBD and control patients. We could not limit ourselves to concrete parts of the colon due to the lack of availability of human tissue. The collection of adequate human material was also difficult because we aimed to collect paired specimens from inflamed and non-inflamed tissue. In the revised version of our manuscript, we have amended the Result section and legend of Figs. 1 and 3, as well as the Material and Methods section, which now reads:
“Paired surgical resections (non-inflamed and inflamed) were taken from the colon of CD and UC patients within the Swiss IBD Cohort study. Non-inflamed colon resections from non-IBD patients undergoing surgery served as our controls.”
- Is there any evidence that the pH is acidic in intestinal inflammation in IBD?
Response: There is a growing body of literature evidencing the association between acidic pH and intestinal inflammation. We believe this is an important point and consequently, we have amended the Introduction section, which now reads:
“Pathologic conditions where localized acidosis may develop include […] inflammatory bowel disease (IBD)[1, 2]. The association between extracellular tissue acidification and intestinal inflammation is ascribed to increased metabolic demand from in-filtrating immune cells, which leads to enhanced glucose consumption and lactic acid formation[3-6]. Additionally, hypoxia, a common feature in intestinal inflammation has been shown to decrease the local pH in the mucosal tissue[7]. Consequently, several studies have demonstrated decreased colonic pH in CD and UC patients compared with healthy subjects[8-11]. During severe inflammation, reduced mucosal bicarbonate secretion, increased bacterial lactate production, and decreased short chain fatty acids might also contribute to the low pH in IBD patient’s colon[12, 13].”
- OGR1 activity may increase “wound healing” and barrier function but may also be detrimental in IBD patients in regard to fibrosis. The reader is somehow left alone with the statement that OGR1 could be a therapeutical target. Would it be wise to inhibit or even stimulate OGR1?
Response: We fully agree that the therapeutic approach involving OGR1 was not fully explored and left undefined. Although our previous animal experiments strongly suggest that the absence of OGR1 has an overall anti-inflammatory effect, the ambivalent role of OGR1 cast doubts on its use as a therapeutic target for IBD. Further research is needed to further understand the overall outcome that inhibiting OGR1 might have and under which conditions could be considered. We have amended the discussion to address this issue, which now reads:
“Although our previous animal experiments[24,40] strongly suggest that the absence of OGR1 has an overall anti-inflammatory effect, the seemingly ambivalent effects of OGR1 inhibition on the gut mucosa promoting reducing inflammation and fibrotic processes while also reducing wound healing responses might indeed preclude its use as a therapeutic target for IBD. Further research is needed to further understand the overall outcome that manipulating OGR1 expression and/or activation might have in IBD patients and under which conditions and in which patient population OGR1 inhibitors could be used.”
and the Conclusions, which now reads:
“OGR1 expression and activity significantly increases with IBD disease activity in IBD-relevant cell types, suggesting an active role of OGR1 in the pathogenesis of IBD. Consequently, OGR1 expression and/or activity could constitute a valuable biomarker for IBD as it correlates with disease activity. Moreover, manipulating OGR1 activation could constitute a novel therapeutic approach for IBD.”
- Increased OGR1 activity was detected in PMBCs from IBD patients. This might be somehow unexpected since this would indicate a systemic effect and not a local one restricted to the intestine. The authors might want to add a comment in that regard.
Response: This was indeed an unexpected result. Further research is needed to further understand the clinical relevance of this finding. We have amended the Discussion section, which now reads:
“Increased OGR1 activity in PBMCs suggests a systemic effect rather than one restricted to the intestinal mucosa. We can only speculate that inflammation in the gut might trigger the expression of OGR1 in circulating PBMCs, although further research is needed to confirm this result and to further understand the clinical relevance of this finding”.
- What could be the mechanisms of increased OGR1 activity in response to acidosis in IBD patients? Is this solely increased gene expression? The authors might want to speculate on that.
Response: We agree with the reviewer in that this is an important discussion point. As a consequence, we have amended the Discussion section, which now reads:
“Earlier work has established that OGR1 can trigger calcium release from intracellular stores, activate protein kinase C, stimulate serum response factor activity and ER stress[14, 18, 27, 53]. Consequently, we hypothesize that OGR1 activity not only leads to an increased OGR1-mediated pro-inflammatory gene expression but also triggers the activation of signalling cascades involved in further pro-inflammatory responses.”
We would like to thank you for the opportunity to resubmit a revised manuscript. We look forward to hearing from you and thank you for your consideration.
Sincerely,
Cheryl de Valliere on behalf of the authors.
Reviewer 2 Report
Attached

Author Response
Dear Reviewer,
Thank you for your comments, which have highlighted some important areas for improvement of the manuscript. We have considered and addressed all your comments and suggestions, as detailed below. We are convinced the manuscript has been improved and trust that our responses fully satisfy your requests.
- Ovarian cancer G-protein coupled receptor 1 (OGR1, also known as GPR68) seems to be “the main actor” of this report; however, it is not always reported with the same diction and as full-text even in the text and the abstract. This may generate a bit of confusion in the reader also for the large number of abbreviations in the text. Therefore, uniformity of nomenclature could improve the clarity of the message of this study.
Response: We thank the reviewer for pointing this out. We have revised the manuscript to ensure that OGR1 (GPR68) is used consistently throughout the text, and we have spelled out the abbreviation in the Abstract. Additionally, we have deleted unnecessary abbreviations (i.e., UPR, GWAS) for readability purposes.
- The pattern of inflammation and fibrosis is different in Crohn’s disease and ulcerative colitis. Do the results of this study reflect these differences? A comment is advisable.
Response: This is indeed a very interesting point. Unfortunately, the limited number of CD and UC samples did not allow us to answer the question adequately as our data don’t reflect the differences between both pathologies. Further studies with a larger number of samples are needed to confirm and further investigate these differences between UC and CD. We have amended the Discussion section to address this limitation of our study, which now reads:
“Additionally, the limited number of samples did not allow us to discern the differences between the inflammatory and fibrotic patterns present in UC and CD, and further studies with a larger sample size are needed to further investigate these differences.”
- Is there a reason why the authors chosed an animal model with a healthy gut and not with inflammatory bowel disease?
Response: We agree that primary fibroblasts isolated from a mouse model of IBD (e.g., the DSS mouse model of colitis) would have potentially shed light on the role of OGR1 in these cells in the context of inflammation. Unfortunately, our previous attempts to isolate fibroblasts from DSS-treated mice have been unsuccessful. In addition, there is a broad consensus, confirmed by our own experience, that the phenotype of primary fibroblasts is lost after around 2 months. This is roughly the time isolation of primary fibroblasts from tissue blocks usually takes and consequently, we did not expose mice to DSS prior to fibroblast isolation.
- “Conclusions: OGR1 expression and activity significantly increase with IBD disease activity, suggesting an active role of OGR1 in the pathogenesis of IBD”. This sentence is too general and renders the results of the study an end in itself and devoid of any possible clinical meaning. A more in-depth analysis of this aspect would be opportune in order to enhance the quality of the subject of the study.
Response: We would like to thank the reviewer for this note and deeper consideration has been given to the clinical implications of our findings. To reflect this, we have amended the Conclusions, which now reads:
“OGR1 expression and activity significantly increase with IBD disease activity in IBD-relevant cell types, suggesting an active role of OGR1 in the pathogenesis of IBD. Consequently, OGR1 expression and/or activity could constitute a valuable biomarker for IBD as it correlates with disease activity. Moreover, manipulating OGR1 activation could constitute a novel therapeutic approach for IBD.”
We would like to thank you for the opportunity to resubmit a revised manuscript. We look forward to hearing from you and thank you for your consideration.
Cheryl de Valliere on behalf of the authors.